# Learning to Compare Nodes in Branch and Bound with Graph Neural Networks

**Abdel Ghani Labassi**
Johns Hopkins University
alabass1@jhu.edu

**Didier Chételat**
Polytechnique Montréal
didier.chetelat@polymtl.ca

**Andrea Lodi**
Cornell University
andrea.lodi@cornell.edu

## Abstract

Branch-and-bound approaches in integer programming require ordering portions of the space to explore next, a problem known as node comparison. We propose a new siamese graph neural network model to tackle this problem, where the nodes are represented as bipartite graphs with attributes. Similar to prior work, we train our model to imitate a diving oracle that plunges towards the optimal solution. We evaluate our method by solving the instances in a plain framework where the nodes are explored according to their rank. On three NP-hard benchmarks chosen to be particularly primal-difficult, our approach leads to faster solving and smaller branch-and-bound trees than the default ranking function of the open-source solver SCIP, as well as competing machine learning methods. Moreover, these results generalize to instances larger than used for training. Code for reproducing the experiments can be found at https://github.com/ds4dm/learn2comparenodes.

## 1 Introduction

Mixed-integer linear programming is an optimization paradigm with applications as varied as airline scheduling [4], CPU management [26], auction design [1] and industrial process scheduling [14]. Modern solvers rely on the branch-and-bound (B&B) algorithm, which recursively divides the search space into a tree, solving relaxations of the problem until an integral solution is found and proven optimal [25]. Throughout this procedure, numerous decisions must be repeatedly made, such as the choice of the variable on which to branch or the choice of primal heuristics to run at every node. These decisions often dramatically impact final performance yet are still poorly understood [3]. Traditionally, these would be made according to hard-coded expert heuristics implemented in solvers. Recently, however, there has been a surge of interest in using machine learning methods to learn such heuristics [5], in particular for variable selection [17, 19, 36, 27, 13].

Despite this success, other critical branch-and-bound decision tasks remain poorly studied. One of the most important is the node comparison problem. Throughout solving, the algorithm must repeatedly select the next node to subdivide, a task known as node selection. It maintains a priority list of the open nodes, ordered according to a node comparison function. This list is then used to select the next node to subdivide, either by simply choosing the highest-ranked node or through some more complex paradigm. Interestingly, a few works have proposed to use machine learning methods to derive node comparison functions [22, 33, 35]. This is particularly promising since the problem is naturally amenable to statistical learning methods. However, despite promising results, challenges hinder progress in this area. Most prominently, it is unclear how to represent nodes, which can vary in the number of variables and constraints. Existing approaches have so far relied on fixed-dimensional representations that necessarily lose information.

In this paper, inspired by work on the related problem of variable selection in branch and bound [17, 19, 36, 27], we propose to tackle this problem by an approach based on graph neural networks (GNNs) [18]. We represent nodes by bipartite graphs with attributes and use a siamese architecture

36th Conference on Neural Information Processing Systems (NeurIPS 2022).

to model the node comparison function. This node representation allows complete information regarding the nodes to be provided to the model, reducing the amount of manual feature engineering. In line with previous work on this node comparison problem [22, 33, 35], we train the network using imitation learning to approximate a diving oracle that plunges towards the optimal solution.

We compare our GNN approach against the support vector machine approach of He et al. [22], the feedforward neural network approach of Song et al. [33] and Yilmaz and Yorke-Smith [35], and the default node selection rule of the open-source solver SCIP [16]. In addition, we compare against the node comparator of this same branching rule but with a highest-rank node selection rule. Results show that our approach leads to improved node selection compared to competing machine learning approaches and, in fact, often improves on the default rule in SCIP itself. In addition, these results generalize to instances larger than those used for training.

The paper is divided as follows. In Section 2, we review the related literature, while in Section 3, we describe the branch-and-bound algorithm and the node comparison problem. In Section 4, we describe our state representation, neural network architecture, as well as training procedure. Finally, we detail experimental results in Section 5.

## 2 Related works

The first steps towards learning node comparison heuristics in branch and bound were taken by He et al. [22]. In this work, they propose to train a support vector machine (SVM) model using the DAGGER algorithm [32] to imitate the node comparison operator of a diving oracle. However, they only use it in combination with a learned pruning model, which cuts off unpromising branches of the branch-and-bound tree, yielding something more analogous to a primal heuristic. They report improvements in the optimality gap against SCIP under a node limit and Gurobi [20] under a time limit on four benchmarks.

Subsequently, Song et al. [33] trained a multilayer perceptron (MLP) RankNet model to perform node comparison using a novel approach they call retrospective imitation learning. In this approach, as applied to the branch-and-bound algorithm, a solver is run until a certain node limit (or potentially until optimality). The node selection trajectory is then corrected into a shortest path to the best solution found during the process. When the solver is run until optimality, this is in effect identical to trajectories generated by the diving oracle. In practice, they generated trajectories using Gurobi and trained using the DAGGER and SMILe [31] imitation learning algorithms. Unlike He et al., they provided results that only use the learned node comparator without an additional pruning operator. On a collection of path planning integer programs, they report impressive improvements in the optimality gap under a node limit against Gurobi and SCIP. However, their appendix also reports more mitigated results on a more challenging combinatorial auctions benchmark used by He et al.

Finally, and more recently, Yilmaz and Yorke-Smith [35] proposed to learn a limited form of feedforward neural network node comparison operator that decides whether the branch-and-bound algorithm should expand the left child, right child or both children of a node. This operator can then be combined with a backtracking algorithm to provide a full node selection policy: in effect, this can be interpreted by combining the neural network node comparator of Song et al. with a node selection rule that only calls it on children of the current node, and reverts to depth-first search otherwise. They use the state encoding from Gasse et al. [17] and train their model using behavioral cloning [30] to imitate an oracle that prioritizes nodes on a path towards one of the $k \geq 1$ best solutions - in effect a generalization of the He et al. oracle. On three benchmarks, they report improvements in time and number of nodes against He et al., and sometimes in nodes against SCIP; in a fourth, they are slightly worse than He et al.

## 3 Background

A mixed-integer linear program (MILP) is an optimization problem of the form

$$\underset{x \in \mathbb{Z}^k \times \mathbb{R}^{n-k}}{\arg\min} \{c^t x : Ax \geq b\},$$

for a matrix $A \in \mathbb{R}^{m \times n}$ and vectors $b \in \mathbb{R}^m, c \in \mathbb{R}^n$. The branch-and-bound algorithm solves this problem recursively as follows. First, the linear program (LP) relaxation $\arg\min_{x \in \mathbb{R}^n} \{c^t x : Ax \geq b\}$

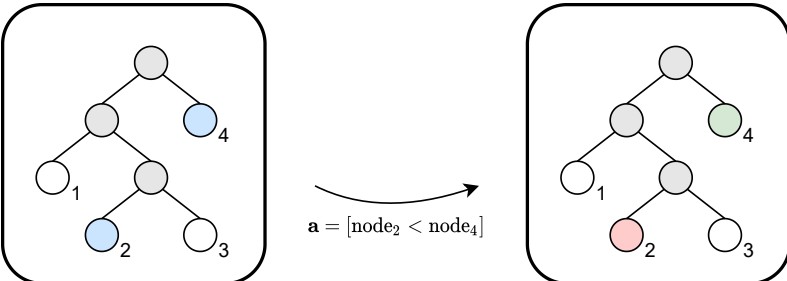

Figure 1: The node comparison problem. Here the solver is asking the NODECOMP function to rank the open nodes 2 and 4, which chose to prioritize the latter over the former.

is solved, which can be done efficiently in practice. This relaxation yields a solution $x^*$, with a lower bound $c^t x^*$ to the MILP. If the LP solution satisfies the integrality constraints, $x^* \in \mathbb{Z}^k \times \mathbb{R}^{n-k}$, the problem is solved. Otherwise, we can take any non-integer $x_i^*$ for some $1 \leq i \leq k$, and divide the problem into two subproblems

$$\arg\min_{x \in \mathbb{Z}^k \times \mathbb{R}^{n-k}} \{c^t x : Ax \geq b, x_i \leq \lfloor x_i^* \rfloor\}, \qquad \arg\min_{x \in \mathbb{Z}^k \times \mathbb{R}^{n-k}} \{c^t x : Ax \geq b, x_i \geq \lfloor x_i^* \rfloor + 1\}$$

The process then starts again, recursively constructing a tree of subproblems with their associated linear relaxation solutions. The branching stops when subproblems are found unfeasible or when their linear relaxations are integral, in which case they furnish feasible solutions. These solutions can be used to prune parts of the branching tree, whose dual bounds are worse than the best-found solution so far.

Throughout this algorithm, nodes in the branch-and-bound tree, corresponding to subproblems, must be selected for further branching: this is known as the node selection problem. In SCIP, this is implemented through a NODESELECT method that takes as argument the list of open nodes and must choose one for subdivision. In practice, it is expensive to rank open nodes at every node selection step, and solvers maintain a priority list of open nodes throughout solving. Whenever new nodes are created, they are inserted in the priority list according to a node comparison function NODECOMP, which takes two nodes as argument and returns whether the first node, the second node or none are to be preferred. The NODESELECT function can then make use of the ranking; in the simplest strategy, it simply selects the node with the highest rank. More complex node selection strategies are also possible, such as prioritizing the highest-ranked children or sibling of the currently opened node over arbitrary leaves. Although this description uses SCIP terminology, other solvers work similarly.

The current state-of-the-art NODECOMP rule, used by default in most solvers, is best estimate search [6, 15]. In this scheme, every node is associated with an estimate of the increase in objective value resulting from selecting the node, computed from pseudocost statistics. The heuristic then selects the node with the highest estimate. Other popular rules include best-first search [21], which prioritizes nodes with the best dual bound, and depth-first search [12], which prioritizes the deepest node.

As detailed by He et al. [22], the task of designing a good NODECOMP function can be assimilated to finding a good policy in a Markov decision process. In this process, the solver is interpreted as the environment, which calls the NODECOMP($\text{node}_1, \text{node}_2$) policy whenever it needs two open nodes compared. This policy is provided information about nodes, which can be interpreted as a state $\mathbf{s} = (\text{node}_1, \text{node}_2)$, and then takes an action as to whether to prefer the first node, the second node, or none, $\mathbf{a} \in \{\text{node}_1\text{-better}, \text{node}_2\text{-better}, \text{equal}\}$. This repeated decision making continues until no new nodes need insertion, that is, until the solving is complete. The process is illustrated in Figure 1.

## 4 Methodology

We now describe our approach to learning good NODECOMP functions. Since the problem can be assimilated to a Markov decision process, we follow previous work [22, 33, 35] and train by imitation learning to mimic an expert policy.

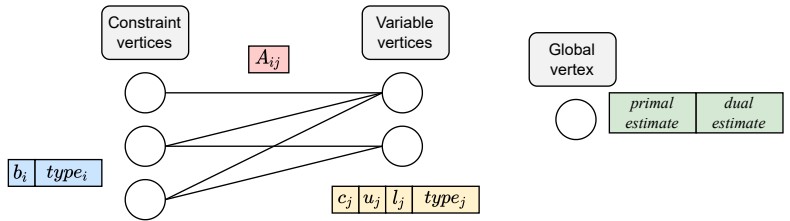

Figure 2: Bipartite graph representation of a node.

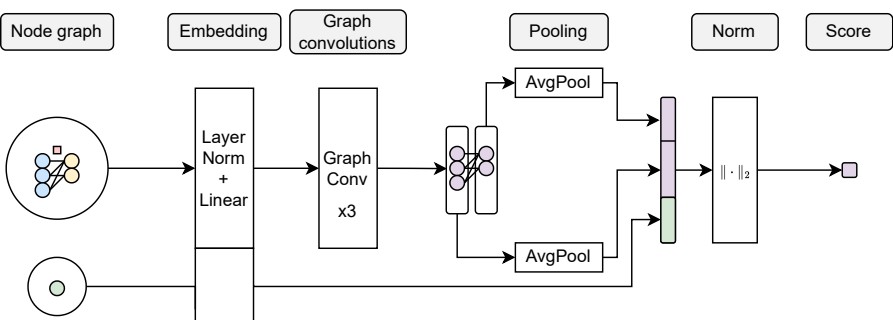

Figure 3: Architecture of the GNN scoring function $g$.

## 4.1 State representation

Our learned NODECOMP($\text{node}_1, \text{node}_2$) takes as input a state $\mathbf{s} = (\text{node}_1, \text{node}_2)$, which represents a pair of nodes. In the related problem of variable selection, important advances were achieved by the usage of bipartite graph representations of the state of the solver, allowing for the first time machine learning to improve over human-designed heuristics in full-fledged solvers [17, 19, 36, 27]. These representations form the current state-of-the-art approach on this problem, and offer many advantages, such as being permutation-invariant in the labeling of the variables and constraints. Inspired by this innovation, we similarly propose to represent each node as a bipartite graph, where on one side there are as many vertices as constraints, and on the other side as many vertices as variables, in the sub-MILP encoded by the node. We draw an edge between a constraint and a variable vertex if the coefficient associated with the variable in the constraint is nonzero. To each constraint vertex $i$ we associate a vector of features, namely its right-hand side $b_i$ and its type ($>, <$ or $=$). Similarly, to each variable vertex $j$ we associate a vector of features, namely its objective coefficient $c_j$, upper and lower bounds $u_j$ and $l_j$, and type (binary, integer, or continuous). In addition, we associate the nonzero coefficient of the variable in the constraint to each edge. Finally, an additional global vertex of attributes associated with the whole node is added, unconnected with the rest. To this vertex, we associate two features, namely an estimate of the objective value of the best feasible solution in the subtree of the node and an estimate of the dual bound achieved at the node, through the `SCIPnodeGetEstimate` and `SCIPnodeGetLowerbound` SCIP functions, respectively. The representation is illustrated in Figure 2.

## 4.2 Model

Our NODECOMP function has the form

$$\text{NODECOMP}(\text{node}_1, \text{node}_2) = \begin{cases} \text{node}_1\text{-better} & \text{if } f(\text{node}_1, \text{node}_2) \leq 0.5, \\ \text{node}_2\text{-better} & \text{if } f(\text{node}_1, \text{node}_2) > 0.5, \end{cases} \tag{1}$$

where $f \in [0, 1]$ is a classification machine learning model. Our model always prefers one node over the other, and never returns $\mathbf{a} =$ equal as an action. The classification model takes the form $f(\text{node}_1, \text{node}_2) = \sigma\big(g(\text{node}_1) - g(\text{node}_2)\big)$ where $\sigma$ stands for the sigmoid function, and $g \in \mathbb{R}$ is a scoring function with a single dimensional, real-valued output. This siamese architecture [7] is naturally symmetric, in the sense that our model satisfies $f(\text{node}_2, \text{node}_1) = 1 - f(\text{node}_1, \text{node}_2)$.

Just as for the design of the state representation, we take inspiration from the latest line of work on variable selection [17, 19, 36, 27] and use a graph neural network [18] model for our scoring function $g$, with suitable modifications for this node comparison problem. A diagram of the architecture is provided as Figure 3. An important advantage of this model is that, just as in the variable selection setting, the same model can be trained on problems of varying number of variables and constraints. In detail, the constraint and variable features of the node are first transformed by an 32-dimensional embedding layer and then pass through three graph convolutional layers, with 8, 4 and 4 dimensions each. Each layer uses a ReLU activation function. The representations of the constraint and variable vectors are then pooled by average separately and then concatenated with the global features of the node. Finally, the resulting vector's $\ell_2$ norm is taken, which is outputted as the score.

## 4.3 Training procedure

Our training procedure is similar to the one of He et al. [22]. Just like them, we aim to imitate a "diving oracle" NODECOMP policy, which prioritizes a node if it contains the optimal solution $x^*$, and falls back on another heuristic (we use best estimate search) if this is not the case:

$$\text{ORACLE-NODECOMP}(\text{node}_1, \text{node}_2) = \begin{cases} \text{node}_1\text{-better} & \text{if } x^* \in \text{node}_1, \\ \text{node}_2\text{-better} & \text{if } x^* \in \text{node}_2, \\ \text{ESTIMATE-NODECOMP}(\text{node}_1, \text{node}_2) & \text{otherwise.} \end{cases}$$

Since nodes represent a partition of the feasible space, the optimal solution cannot be in the feasible spaces of both nodes simultaneously, so this is well-defined. As this NODECOMP function uses knowledge of the optimal solution, it cannot be used in practice; however, it can be run on training instances by precomputing optimal solutions, and it is worthwhile to try to imitate its decisions without this additional knowledge. To do this, He et al. use DAGGER, an expensive imitation learning algorithm that aims to diversify the states from which the expert is sampled through several rounds of training. We propose a simpler procedure that achieves a similar result with lower computing requirements.

This procedure runs as follows. We first solve the instances using a solver, collecting their optimal solutions. We then solve the instances again, using a plain highest-priority NODESELECT rule. When the solver calls the NODECOMP function, we query the oracle, and if it chooses node$_1$-better or node$_2$-better, we collect state information $\mathbf{s}$ and the resulting decision $\mathbf{a}$ as an expert sample $(\mathbf{s}_i, \mathbf{a}_i)$. Next, crucially, we take the opposite decision than the oracle recommends, making a mistake on purpose. This increases the variety of states explored during the sampling phase and makes the state distribution more aligned with the machine learning policy, which is bound to make mistakes. We follow this procedure until the solving is completed.

As a result of this sampling process, we obtain a dataset of expert samples $\mathcal{D} = \{(\mathbf{s}_i, \mathbf{a}_i)\}$ from which to train our machine learning policy. Since we saved samples when the oracle had a preference, the actions can be interpreted as labels 0 or 1 according to whether the first or second node was preferred. Learning the preference of the oracle then becomes a simple classification task that can be performed by minimizing a cross-entropy loss over our classifier $f$. Since mistakes coming early on in the sampling process can be exponentially costly, we weight the samples during training using an exponentially decreasing scheme, $w = \exp(1 + |d_1 - d_2|)/\min(d_1, d_2)$, where $d_1, d_2$ are the depths of the first and second nodes, respectively. This is similar to the exponential weighting scheme used by He et al.

## 5 Experimental results

We now present experimental results on three NP-hard problems. We evaluate each machine learning method by running it in SCIP with a simple highest-priority rule that falls back to ESTIMATE after two feasible solutions have been found, as we detail in Section 5.4. We also evaluate the default SCIP node selection rule (that is, with both default NODESELECT and NODECOMP). Code for reproducing these experiments can be found at https://github.com/ds4dm/learn2comparenodes.

## 5.1 Benchmarks

We evaluate on three NP-hard instance families that are particularly primal-difficult, that is, for which finding feasible solutions is the main challenge. Those are instances for which improved

Table 1: Test accuracies of the different machine learning methods in imitating the diving oracle.

| | Test FCMCNF | Test MAXSAT | Test GISP |
|---|---|---|---|
| SVM | 91.5% | 90.6% | 93.0% |
| MLP | **97.8%** | **97.9%** | 95.6% |
| GNN | 95.7% | 97.7% | **97.0%** |

node comparison is likely to have a particularly broad impact, so differences between methods should be clearer. The first benchmark is composed of Fixed Charge Multicommodity Network Flow (FCMCNF) [23] instances, generated from the code of Chmiela et al. [10]. We train and test on instances with $n = 15$ nodes and $m = 1.5 \cdot n$ commodities, and also evaluate on larger transfer instances with $n = 20$ nodes. The second benchmark is composed of Maximum Satisfiability (MAXSAT) instances, generated following the scheme of Béjar et al. [9]. We train and test on instances with a uniformly sampled number of nodes $n \in [60, 70]$ and transfer on instances with $n \in [80 - 100]$. Finally, our third benchmark is composed of Generalized Independent Set (GISP) [11] instances, generated from the code of Chmiela et al. [10]. We train and test on instances with a uniformly sampled number of nodes $n \in [60, 70]$ and transfer on instances with $n \in [70 - 80]$. All these families require an underlying graph: we use in each case Erdős-Rényi random graphs with the prescribed number of nodes, with edge probability $p = 0.3$ for FCMCNF and $p = 0.6$ for MAXSAT and GISP.

## 5.2  Baselines

We compare against the state-of-the-art best estimate node comparison rule [6, 15]. This is the NODECOMP function used by default in SCIP, in conjunction with a diving NODESELECT rule that prioritizes children and siblings of the currently focused node. To disentangle the effect of this NODESELECT rule, we report both results with this rule (default SCIP) and with a plain NODESELECT that always selects the highest-ranked node (ESTIMATE). We also report the performance of the expert we aim to imitate, the diving oracle (ORACLE). This method cheats by having access to the optimal solution ahead of the solving.

In addition, we compare against two competing machine learning approaches: the support vector machine [34] approach of He et al. [22] (SVM) and the RankNet feedforward neural network [8] approach of Song et al. [33] and Yilmaz and Yorke-Smith [35]. The former uses a multilayer perceptron; the latter uses the same, except for one benchmark where they use three hidden layers. For simplicity, we use a multilayer perceptron for all benchmarks (MLP), with a hidden layer of 32 neurons. The features used in the three papers are roughly similar; again, for simplicity, we use the fixed-dimensional features of He et al. for both the SVM and the MLP. All methods except the default SCIP use a plain highest-rank NODESELECT.

## 5.3  Training

We use the training procedure of Section 4.3 for all machine learning models. The SVM model is trained using the scikit-learn [29] library; the MLP and the GNN implemented in PyTorch [28] and optimized using Adam [24] with training batch size of 16. Running the sampling procedure on 1000 training and 100 test instances yielded 16285 training and 3019 test samples for FCMCNF, 41299 training, and 4868 test samples for MAXSAT, and 41299 training and 4868 test samples for GISP. We train/evaluate using an Nvidia® Tesla V100 GPU and an Intel® Xeon Gold 6126 CPU. Test accuracies of the different models can be found in Table 1.

## 5.4  Evaluation

We evaluate each machine learning model by setting SCIP's NODECOMP function be the NODECOMP function associated with the machine learning model (Eq. 1). In parallel, the NODESELECT function is designed to use the highest ranked node according to this node comparison function, until two feasible solutions have been obtained. After this state, it falls back to ESTIMATE. This has the effect of prioritizing the learned node comparison during the initial phases of the solving, in a size-independent manner.

Table 2: Evaluation of node comparison methods in terms of the 1-shifted geometric mean of the number of nodes and solving time (in seconds) over the instances, with the geometric standard deviation. For each problem, machine learning models are trained on instances of the same size as the test instances, and evaluated on those and the larger transfer instances (50 instances each).

|  | Test FCMCNF | | Transfer FCMCNF | | Test MAXSAT | | Transfer MAXSAT | | Test GISP | | Transfer GISP | |
|---|---|---|---|---|---|---|---|---|---|---|---|---|
|  | Nodes | Time | Nodes | Time | Nodes | Time | Nodes | Time | Nodes | Time | Nodes | Time |
| ORACLE | 15±4 | 3.80±1.5 | 75±4 | 19.9±1.8 | 102±2 | 6.17±1.8 | 160±2 | 8.9±1.5 | 98±3 | 4.18±1.3 | 1062±2 | 22.6±1.5 |
| SCIP | 41±5 | 4.64±1.5 | 178±4 | 26.7±1.9 | 147±2 | 9.26±1.5 | 171±2 | 12.9±1.4 | 184±2 | **4.38**±1.2 | 1533±2 | **19.1**±1.5 |
| ESTIMATE | 21±5 | **4.09**±1.5 | 122±5 | **23.8**±2.0 | 177±2 | 8.16±1.7 | 247±2 | 12.1±1.6 | 218±2 | 4.64±1.3 | 1435±2 | 24.9±1.7 |
| SVM | 20±5 | 4.10±1.5 | 133±5 | 24.8±1.9 | 150±3 | 7.34±1.8 | 225±2 | 10.7±1.6 | 207±3 | 4.57±1.3 | 1295±2 | 23.4±1.6 |
| MLP | 21±5 | 4.15±1.5 | **115**±5 | 24.1±1.9 | 157±3 | 7.76±1.9 | 215±2 | 10.8±1.6 | 209±3 | 4.72±1.3 | 1238±2 | 23.0±1.6 |
| GNN | **19**±5 | 4.14±1.5 | 122±5 | 24.5±1.9 | **117**±3 | **6.66**±1.9 | **171**±2 | 9.1±1.6 | **170**±3 | 4.64±1.3 | **1203**±2 | 22.8±1.5 |

We average results over the benchmarks using the 1-shifted geometric mean with geometric standard deviation to measure the average and dispersion of B&B tree size and solving time on our benchmarks. This metric is the standard used in the mixed-integer programming community since it reduces outlier effects from both directions (too easy and too hard instances), as discussed in Appendix A3 of Achterberg [2]. We evaluated on 50 test and 50 transfer instances, as explained in Section 5.1. Table 2 summarizes the results.

### 5.5 Discussion

As can be seen in Table 1, both the MLP and GNN achieve similar accuracies on the datasets, with the SVM lagging a bit more behind. When used in solving, however, the GNN more consistently dominates the other machine learning approaches. More impressively, the model is often competitive with or even better than the SCIP default node strategy, particularly on the MAXSAT problems. In addition, these results generalize to larger instances than those trained on. This is the case despite using a plain NODESELECT rule, which suggests that most of the difficulty in node selection can be reduced by the design of a good NODECOMP function. This is particularly attractive as this is a problem that is naturally amenable to machine learning methods, as described in this work.

A disadvantage of the imitation learning approach we follow is that it is limited by the performance of the expert itself. If the oracle does not beat a baseline, imitating it is unlikely to bring gains. A good example is the largest benchmark, transfer GISP: this is the only family where the oracle does not beat the SCIP default rule in time. Therefore, it is unsurprising that no other machine learning method was able to beat it. Note that nonetheless, the GNN is the model that manages to come the closest to the performance of the oracle on this benchmark, suggesting strong imitation capabilities.

## 6 Conclusion

This work proposes to train a graph neural network to compare nodes in a branch-and-bound solver for solving mixed-integer linear programs. We represent nodes as bipartite graphs with features and train a neural network to imitate a diving oracle that plunges towards the optimal solution. On three primal-difficult NP-hard benchmarks, our approach outperforms prior machine learning approaches and often even the SCIP default node selection strategy, while generalizing to larger instances than trained on.

An interesting direction for future work would be to combine variable and node selection strategies. Besides the fact that the two problems are tightly linked, good node selection is particularly important in primal-difficult problems, while good branching is particularly useful in dual-difficult problems. Combining the two could thus help outperform current expert-designed strategies on generic problems.

### Acknowledgements

This work was supported by the Canada Excellence Research Chair program. In addition, the first author was affiliated with the Department of Computer Science and Operations Research of the University of Montreal during most of the project.

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
