# OpenReview forum: "Learning to Compare Nodes in Branch and Bound with Graph Neural Networks"
_NeurIPS.cc/2022/Conference — NeurIPS 2022 Accept_

### Official Review · Reviewer_zJfP · 2022-07-09

**Rating:** 3
**Confidence:** 3
**Soundness:** 3 good
**Presentation:** 1 poor
**Contribution:** 1 poor

**Summary:**

This paper proposes a variable selection method for MILPs based on GNNs, diving oracle, and imitation learning.

**Questions:**

- What is the most significant contribution of this paper that the authors want to inform the NeurIPS community?
- What are the qualitative differences against the existing GNN-based approaches for MIP variable selection?

**Strengths And Weaknesses:**

# Strengths

- The proposed method outperforms the SCIP solver in the experiments.
- Three hard problems with standard and transfer settings are used in the evaluations.

# Weaknesses

- The proposed method is a combination of known techniques and provides few technical insights. This paper does not reach the NeurIPS standards of technical contributions.
- The proposed approach looks very similar to Solving Mixed Integer Programs Using Neural Networks (Nair et al., 2021). Please discuss the differences and qualitative merits of the proposed method and compare them in the experiments.
- There are many other ML methods for MILPs, e.g., learn2branch (Gasse et al., 2019) and many others cited in this submission. Please discuss the qualitative merits of the proposed method and compare them in the experiments.

Overall, I feel that this paper fails to contextualize itself. More discussions and evaluations are needed to demonstrate the effectiveness of the proposed method over other approaches.

---

> ### Author Response · Authors · 2022-08-02
> **Response to Reviewer zJfP**
>
> We thank the reviewer for their time and insights in reviewing our paper. However, we believe there is a misunderstanding with regards to what the paper is achieving. There are several distinct decision making tasks that are repeatedly performed by solvers when solving MILPs. One is the task of selecting which node (sub-MILP) in the B&B tree to expand next, known as the node selection problem. Another problem is, given the node selected by the node selection rule, to select which variable to branch on. This is the variable selection problem. Although of course every piece in the algorithm interacts with each other in complex ways, those two tasks are nonetheless distinct, are treated by different APIs in solvers, and do not have the same action set (open nodes for the first one, variables of a node in the second).
>
> This work attacks the node selection problem, not the variable selection problem, and we do so by learning a GNN-based preference function that orders pairs of nodes. Thus the statement  “[t]his paper proposes a variable selection method” is inaccurate, and comparing against variable selection papers such as Gasse et al. (2019) or Nair et al. (2021) does not make sense. However, we do take inspiration from their works, namely in their successful usage of a bipartite graph representation and GNN models, to design an approach to this related, but different problem. But one cannot use the models of one problem to the other problem. The inputs and outputs are not even the same: for variable selection, the input is a bipartite graph representing a node, and the output is a probability distribution over the variables {x_1, …, x_n}, of that node, while in the node comparison problem, one is given a pair of bipartite graphs, and one must return -1 or 1 (or a probability).
>
> Thus there appears to be a misunderstanding with respect to the goals of the paper, and we propose to better highlight the connection with works on variable selection, and how they can help guide new node selection methods, while being careful to distinguish these works. But we also beseech the reviewer to take a second look at the paper, as we believe we had spent considerable effort in detailing the node selection problem, and judge by themselves as to the value of the proposed method, since we believe that it forms a substantial step forward towards the integration of deep learning methods in MILP solvers.

---

### Official Review · Reviewer_zm9Z · 2022-07-11

**Rating:** 3
**Confidence:** 4
**Soundness:** 3 good
**Presentation:** 2 fair
**Contribution:** 1 poor

**Summary:**

In this paper, the authors design a GNN model for comparing two Branch and Bound (BnB) search nodes of Mixed Integer Linear Programming (MILP) and use this model to replace the node comparison function of an existing MILP solver SCIP.

**Questions:**

See the above weaknesses.

**Limitations:**

The authors discussed that the performance is limited by the performance of the oracle expert.

**Strengths And Weaknesses:**

Strengths:
1.	The idea of using a GNN model to evaluate two BnB search nodes is novel and it has great significance to BnB search in other research fields of combinatorial optimization, such as SAT, Traveling Salesman Problem, Maximum Clique Problem, Graph Coloring Problem, etc.

Weaknesses:
1.	This work uses an antiquated GNN model and method, it seriously impacts the performance of this framework. The baseline algorithms/methods are also antiquated.
2.	The experimental results did not show that this work model obviously outperforms other variant comparison algorithms/models.
3.	The innovations of network architecture design and constraint embedding are rather limited.

---

> ### Author Response · Authors · 2022-08-02
> **Response to Reviewer zm9Z**
>
> We thank the reviewer for their time and comments. We will address each point in the weaknesses section in turn.
>
> Regarding the first point, we concede that the GNN architecture we used does not make use of the most recent mechanisms proposed in the GNN literature, such as attention mechanisms, frequency-targeting filters, or sophisticated pooling techniques. We must admit, however, being puzzled that this is kept as a point of objection: this is the first time anybody proposes any deep learning at all for attacking this problem, let alone graph neural networks. We endeavor to show that using deep learning to attack this problem leads to improvements compared to simpler models such as SVMs and MLPs that had been proposed in the past - which exact GNN convolution, say, is best for this task seems to us best left for future work once the advantages of the approach are already established.
>
> Moreover, regarding the baseline methods being antiquated, He et al. (2014), Song et al. (2018) and Yilmaz and Yorke-Smith (2021)  are the latest (and sole) proposals in the literature! Surely we cannot be faulted for the competitors not having used more complex models than an SVM and a MLP. This is in fact, exactly our proposal: we propose, for the first time, to use deep learning methods, namely with a GNN, and we show that this leads to improvements.
>
> Regarding the second point, with respect to the soundness of the experimental results, we show on every benchmark except “time GISP” improvements over SCIP, and even on the latter we are competitive. Considering how difficult it is to improve over a state-of-the-art solver such as SCIP, we believe that these results already stand out as demonstrating our approach’s value  for the node comparison problem in branch and bound.
>
> Finally, regarding the “innovations of network architecture design and constraint embedding”: the paper is not looking to provide new GNN architecture or constraint embeddings, but show that using a GNN at all is useful for the node comparison problem. We do suggest using a siamese architecture, which has the advantage of making the comparison function symmetrical by design, but of course we did not invent this design, nor do we claim to. Whether this is the optimal choice for this problem is an open question, which would surely lead to a very interesting follow up paper, but as explained to our answer to the first point, our objective was first and foremost to show that deep learning can be useful here at all, and we settled on a reasonable enough design which seems to perform well.

---

### Official Review · Reviewer_ksTx · 2022-07-17

**Rating:** 5
**Confidence:** 3
**Soundness:** 3 good
**Presentation:** 3 good
**Contribution:** 2 fair

**Summary:**

Branch and bound is a fundamental component for solving the mixed integer linear programming problem. As one of the building block, node comparison function takes the responsibility of choosing next node to subdivide. In this work, a graph neural network (GNN) method is proposed by formalizing the way to represent the nodes when searching the next step to go. Different from previous work, the node features are extracted from the associated properties of linear equations and objective function. Oracle node division is regarded as the supervised data to train the scoring function based on GNN model. It's interesting to see the application of GNN method to solve NP hard problems after revisiting the original optimization task. The novelty of this work lies at the employment of GNN to learn the node representations to mimic the ideal branch-and-bound solution.

**Questions:**

Please refer to the weaknesses.

**Limitations:**

yes

**Strengths And Weaknesses:**

Strengths:
1. This work gives alternative way to represent the nodes with extracted features from the linear constraints and objective function. A bipartite graph to show the connection between constraints and variables builds the basics for applying the GNN method to score the node in branch-and-bound when comparing a pair of nodes.
2. As one of the most important part, the proposed method employs imitation learning to optimize the scoring function through learning from the oracle branch-and-bound progress.
3. Thorough experiments are conducted over real NP hard problems to valid how the proposed method works.

Weaknesses:
1. The motivation about the reason why employing GNN works is not clear enough. It's better to further comment on why the formalized bipartite graph works and what kinds of benefits that the other methods can not bring?
2. It's hard to understand how the proposed method can adapt to a new MILP problem. From the presentation of this work, we can see that the training data to learn a node selection policy is based on oracle optimization progress for the problems in hand. One important question that needs to answer is how we can apply the learned GNN-based scoring function to a new problem. After reading the manuscript several times, it's still difficult to catch up with this point.
3. The experimental results seem to be not strong enough to demonstrate the superiority of the proposed. From the results shown in Table 1 and 2, we can see that GNN-based method can beat the baseline in some tasks, but not show absolute advantage. It deserves more words and insight to explain the function of GNN method.

Minor Issue:
1. It's difficult to draw connection between Figure 1 and the description shown from Line 111-117. The illustration only shows the comparison between a pair of nodes, but missing key concepts of Markov Decision Process, such as the state, action, and the environment.

---

> ### Author Response · Authors · 2022-08-02
> **Response to reviewer ksTx**
>
> We thank the reviewer for the thorough review. We will address the issues pointed out in the weakness section in order.
>
> For the first point, regarding the choice of GNNs, we represent nodes using bipartite graph, and correspondingly use a GNN to model the policy, simply because it is the state-of-the-art for variable selection in MILP solving. Although not the same task, it is related and so it was reasonable to expect that it could fruitfully be used for the node selection/comparison task as well. Moreover, the three main motivations that led to this representation being used in variable selection also transfer for node selection/comparison, namely, the representation encodes the entirety of the information about the node (it is a raw representation), it is invariant to permutations of variables and rows, and the same trained model can be used on nodes of differing sizes. We understand however that this motivation might not have been very clear, and we propose to improve its justification at the start of Section 4.
>
> For the second point, we are not sure how to interpret the question. If the question is how the policy is used at inference time, we implement a custom node comparison operator in SCIP, which is a function NODECOMP(node_1, node_2) that SCIP will call whenever it wants to order these two nodes. In our case, at test time, we implement a simple NODECOMP function in the SCIP API that computes, for the two nodes that needs to be compared, their graph bipartite representations, and then call the GNN policy to extract a preference (node 1 or node 2). The NODECOMP function then passes on the GNN policy preference back to SCIP. This GNN uses the weights that have been selected (trained) at training time from the examples from the oracle. This might not have been very clear and we propose to rework section 5.4 to remove any ambiguity.
>
> Another interpretation of your question is more philosophical, as follows. The expert is a node comparison policy that has access to extra information (the optimal solution), but the GNN model must imitate it without having access to this extra information (which is why it can be used at test time.) Then perhaps your question is: why do you even expect the GNN to be able to imitate this oracle well, if it does not have access to the same features (missing this extra information.) On this front, we are not certain either, except that 1) there might be patterns between the optimal plunging patterns and the MILP problem data (A,b,c) that a machine learning model could be able to pick up, and 2) prior work (He al. 2014) has demonstrated that it was possible. We are simply extending their work by using more raw representation, and deep learning models for learning from these more raw representations.
>
> Finally, regarding the third point about the weakness of the experimental results, we respectfully disagree. On every benchmark except “time GISP”, our approach leads to improvements in time with respect to default SCIP, and on this latter benchmark we are quite competitive. This is, we believe, a substantial achievement considering how hard it is to improve over full-fledged MILP solvers, such as SCIP, which are the results of decades of development and research in the operations research community.

---

### Meta-Review · Area_Chair_ZEma · 2022-09-02

**Recommendation:** Accept
**Confidence:** Less certain

**Metareview:**

This paper clearly documents a well-executed exploration of an imitation learning / neural diving approach to improving node selection in BB solvers using a GNN approach.
I believe this paper is a useful contribution that pushes forward the important project of integrating modern ML techniques to improve integer programming.

The reviewers were less sanguine than the meta-reviewer. I will explain why.

The authors note, and I concur, that node selection strategies in modern BB solvers have been *heavily* researched for decades, so beating their performance on the majority of these difficult tasks is quite an achievement! (This point confused the first reviewer.)

The second reviewer was disappointed that the authors used an "antiquated" GNN architecture, or compared to "antiquated" baselines, but did not substantiate which modern architectures or comparisons would have been better. In the absence of constructive criticism, I construe these as a misapplication of the standards of deep learning (very fast progress in beating benchmarks) to integer programming (a mature field where progress is slower).

While several GNN approaches are available for variable selection in BB solvers, node selection is an important and independent challenge.
I am not aware of any previous work on deep learning for node selection (nor are the authors),
which explains why the authors chose to compare their GNN-based approach to other strategies they created and the default strategy in SCIP rather than the (non-existent) deep-learning SOTA for the problem.
 (This point confused the third reviewer.)

**Award:**

No

---

### Decision · Program_Chairs · 2022-09-14

Accept